# Motor Function of Children with SMA1 and SMA2 Depends on the Neck and Trunk Muscle Strength, Deformation of the Spine, and the Range of Motion in the Limb Joints

**DOI:** 10.3390/ijerph18179134

**Published:** 2021-08-30

**Authors:** Agnieszka Stępień, Ewa Gajewska, Witold Rekowski

**Affiliations:** 1Department of Rehabilitation, Józef Piłsudski University of Physical Education, Marymoncka Str., 00-968 Warszawa, Poland; orthosas@wp.pl (A.S.); witold.rekowski@awf.edu.pl (W.R.); 2Department of Developmental Neurology, Poznan University of Medical Sciences, 49 Przybyszewskiego Str, 60-355 Poznan, Poland

**Keywords:** spinal muscular atrophy, motor function, scoliosis, range of motion, physiotherapy, rehabilitation, Hammersmith Functional Motor Scale Expanded (HFMSE), CHOP INTEND

## Abstract

The purpose of this study was to investigate the functional relationships between selected ranges of motion of the neck, upper and lower limbs, the strength of the neck and trunk muscles, postural parameters, and the motor function of children with SMA1 and SMA2—27 children, aged 6 months-15 years, with genetically confirmed spinal muscular atrophy type 1 (19 children) and 2 (8 children) undergoing pharmacological treatment. All children were examined, according to the methodology, including the motor function evaluation, measurement of selected ranges of motion, assessment of postural parameters, and measurement of neck and trunk muscle strength. The functional status of 15 children was assessed with the CHOP INTEND (CHOP group) scale and of 12 children with the HFMSE (HFMSE group). The results obtained showed that, in children examined with the CHOP scale, greater limitation of flexion in the shoulder joints was observed. As the deformation of the chest increased, the functional abilities of children deteriorated. In participants examined with the CHOP group, the ranges of neck rotation decreased with the increase of the chest deformity. In the HFMSE group, the ranges of head rotation showed a strong relationship with some parameters of muscle strength and the sum of the R coefficients. Participants showed many significant relationships between the range of motion in the neck and joints of the limbs, with more significant relationships in the CHOP group. The following conclusions were made: motor skills of children with SMA depend on muscle strength, range of motion, and deformities of the spine and chest; the development of scoliosis adversely affects the motor function, ranges of motion, and muscle strength; and movement ranges are related to motor skills and strength values.

## 1. Introduction

Spinal muscular atrophy (SMA) is a rare genetic disease [1] that leads to many limitations in the daily functioning of the affected persons and their families. The disease has a progressive course, featuring weakened muscle strength [2,3,4,5,6], progressive deformities of the spine and chest [7,8], limitations of the range of motion (ROM) in the cervical spine and limb joints [8,9,10], respiratory disorders, and the gradual deterioration of the functional status [6,11,12,13].

The standards of care for people with SMA recommend the implementation of multi-specialist care. One of the important elements of the process is physiotherapy, which is aimed at improving or maintaining the motor skills of the patients. The planning of the treatment should be based on a detailed examination [14].

In the last few years, new options of pharmacological therapy have appeared. Examinations have confirmed the effectiveness of Nusinersen treatment in patients with type 1 (SMA1), type 2 (SMA2), and type 3 (SMA3) SMA [14,15,16,17,18]. Recently, another drug, Risdiplam, received its first approval in the USA for the treatment of children with SMA, aged 2 months and older [19,20].

The basic element in assessing the effectiveness of treatment, used in persons with SMA, is the examination of the functional status and the trajectory of changes. Scales of proven reliability are used to assess the progress of motor function (MF), including the Children’s Hospital of Philadelphia Infant Test of Neuromuscular Disorders (CHOP INTEND) [21,22] and Hammersmith Functional Motor Scale Expanded (HFMSE) [23,24,25]. CHOP INTEND was designed to assess the MF in weak children and has been validated in children with SMA 1. It consists of the evaluation of 16 functions, where the score can range from 0–64 [21,22].

Over the last dozen or so years, many attempts have been made to develop the best tool for the assessment of gross motor skills in patients with SMA 2 and 3. Hammersmith Functional Motor Scale (HFMS) was developed in 2003 as a clinical and research tool. It is an ordinal scale, consisting of 20 items, each of which is assigned a score from 0–2. Then, this scale was modified and extended with a further 13 items to create the Hammersmith Functional Motor Scale Expanded (HFMSE) [23,24,25].

According to the International Classification of Functioning, Disability and Health (ICF), adopted by the World Health Organization, an examination should include an assessment of individual body structures and functions, as well as activity and participation. In studies involving patients with SMA, attempts have been made to identify factors that may affect the deterioration of the quality of everyday functioning. It has been shown, among others, that there is a connection between the MF and the muscle strength of the limbs [2,4,5,13,26] and selected ranges of motion in the joints of the lower limbs.

The aim of this study was to broaden the knowledge of the structures and functions of the musculoskeletal system that may affect the MF of children with SMA.

The main objective was to investigate the functional relationships between selected ranges of motion of the neck, upper and lower limbs, the strength of the neck and trunk muscles, postural parameters, and the MF of children with SMA1 and SMA2.

## 2. Methods

This study was carried out as part of the “Education in the new reality: a comprehensive and long-term model of physiotherapeutic treatment in spinal muscular atrophy” project, developed by Stowarzyszenie Lwie Serca, financed by the National Freedom Institute, Civic Initiatives Fund Program for 2014–2020, with the consent of the Senate Bioethics Committee at Józef Piłsudski University of Physical Education (SKE 01–28/2019). The project involved a detailed initial assessment of the functional status of each child, identification of the most important therapeutic needs, and a re-examination after a period of 10–12 months. In the presented paper, the data obtained during the initial examination of children qualified for the project were used.

### 2.1. Participants

Children, aged 6 months-15 years, with genetically confirmed spinal muscular atrophy type 1 and 2, undergoing pharmacological treatment with Nusinersen, after genetic treatment or participating in clinical trials of other drugs clinical trials, were enrolled in the study. The number of participants was limited to 27, due to the project financing assumptions. The order of application submissions was decisive for admission.

### 2.2. Examinations of Participants

All children were examined, according to the methodology, including the MF evaluation, measurement of selected ranges of motion, assessment of postural parameters, and measurement of neck and trunk muscle strength.

The CHOP INTEND and HFMSE functional scales were used to assess MF. The CHOP INTEND scale was used to examine children with SMA1. According to the scale, the following movements were assessed: spontaneous movements of upper and lower extremities in supine and sitting; hand grip, head maintaining in midline, and head rotation with visual stimulation in supine; rolling to the side; neck flexion in supine and head/neck extension in ventral suspension [21,22].

In children with type 2 SMA, examined with the HFMSE, quality of motor functions/activities, such as: plinth/chair sitting, long sitting, one/two hands to head in sitting, supine to side lying, rolling prone to supine, rolling supine to prone, sitting to lying, propping on forearms, lifting head from prone, propping on extended arms, lying to sitting, four-point kneeling, crawling, lifting head from supine, supported standing, unsupported standing, stepping, right and left hip flexion in supine, high kneeling to half kneeling, high kneeling to standing, stand to sitting on the floor, squat, jumping forward, and ascending and descending stairs (with railing and without arm support) were assessed [24]. Children with SMA1, who were in good functional condition (able to seat, with high CHOP INTEND score), owing to the implemented pharmacological treatment, were also assessed with the HFMSE scale. 

A physiotherapist, who used the CHOP INTEND and HFMSE scales, had previously been trained and participated in annual reminder meetings. A manual containing all the procedures for applying the scales was used during the assessment. All items were tested without using a brace or orthoses.

All participants were measured with a plurimeter (Rippstein, Switzerland) in the supine position. Ranges of left and right neck rotation, shoulder joint flexion, elbow extension, hip extension, and hip joint flexion with knee extension were assessed. Then, based on the measurements of the ROM, the following parameters were calculated for each participant: sum of cervical rotation to the left and right (CRS), sum of shoulder flexion limitation (LSF) in both shoulders, sum of elbow extension limitation in both elbows (LEE), sum of hip extension limitation (LHE), and sum of hip flexion with knee extension limitation (LHF) in both hips.

The chest shape was also assessed in the supine position using the SATR (Supine Angle of Trunk Rotation) test in the upper (SATRU) and lower (SATRL) parts of the chest. Measurements of the angle of trunk rotation (ART) in the thoracic (ATRT) and lumbar (ATRL) spine in a sitting position, included in the Society on Scoliosis Orthopaedic and Rehabilitation Treatment (SOSORT) guidelines [26] were made; however, due to the significant weakness of muscle strength and difficulty in independently leaning forward, participants’ heads and shoulders were supported during the measurement. In children who were able to sit independently, pelvic obliquity (PO) was also assessed. The ATR, SATR, and PO tests were conducted with the use of a scoliometer (Orthopedic Systems Inc., Union City, CA, USA, OSI 1995). Earlier studies confirmed reliability of the CR, SATR, PO, and HE tests in the group of children with SMA [27].

In all of the children under examination aged 5 and above, measurements of maximum voluntary isometric contraction of muscles flexing and straightening the neck and trunk, in the side-lying position, with the lower limbs at the angle of 45 degrees in the hip and knee joints, were made. A handheld digital muscle tester, MICROFET2 (Hoggan Scientific, LLC, Salt Lake City, UT, USA), was used for the measurements. A dynamometer is used to measure muscle strength in pounds, newtons, or kilograms, with an accuracy of 0.1 pound, in the range of 0.8–300 pounds. Each participant had 12 measurements (6 on each the left and right sides): neck flexion in side lying on the left (NFL), neck flexion in side lying on the right (NFR), neck extension in side lying on the left (NEL), neck extension in side lying on the right (NER), right scapula forward in side lying on the left (SFL), left scapula forward in side lying on the right (SFR), right scapula backward in side lying on the left (SBL), left scapula backward in side lying on the right (SBR), right part of pelvis forward in side lying on the left (PFL), left part of pelvis forward in side lying on the right (PFR), right part of pelvis backward in side lying on the left (PBL), and left part of pelvis backward in side lying on the right (PBR). Due to the possible fatigue of the children, each of the measurements of the strength of the neck and trunk muscles was performed once. Earlier, as part of a warm-up, trial measurements were made on the left and right sides. Force measurements were performed by an experienced physiotherapist (who had previously received training and had experience in measuring force) with a hand dynamometer.

Force measurements [F; in newtons] were multiplied by the moment arm [a; distance in meters], measured from the axis of motion [the forehead center-suprasternal notch for the neck measurements; chest width for the upper trunk measurements, pelvic width for the lower trunk measurements] and divided by body weight [BM; in kilograms], yielding the relative torque R [R = F × a/BM] coefficient for each of the measurements. Then, the sum of the R (RS) coefficients from all measurements was calculated. A similar method of normalization by weight has been used in other studies, in patients with neuromuscular diseases.

Earlier studies demonstrated high reliability of measurements of the neck and trunk muscle strength, performed according to the above methodology, in children with SMA and healthy children [28].

All measurements and their abbreviations are shown in Figure 1.

### 2.3. Statistical Methods

For a statistical analysis, IBM SPSS Statistics v.20 software was used. The mean values and standard deviations for all assessed parameters in the groups of children, adopted due to the applied CHOP and HFMSE functional scales, were calculated. 

Due to the number of participants and the absence of a normal distribution, the Mann-Whitney U test was used to compare the values of the parameters obtained in the groups.

The Spearman correlation was used to analyze the correlation between the MF score, range of motion values (CRS, SFL, EEL, HEL, and HFL), muscle strength measurements (NFL, NFR, NEL, NER, SFL, SFR, SBL, SBR, PFL, PFR, PBL, and PBR), and postural parameters (Cobb angle, ATRT, ATRL, SATRU, SATRL, and PO). The following levels of correlation strength were adopted: <0.3 negligible correlation, 0.3—<0.5 low correlation, 0.5—<0.7 moderate correlation, 0.7—<0.9 high correlation, and 0.9—very high correlation [29]. The level of significance was set at *p* ≤ 0.05.

## 3. Results

A total of 27 children participated in the project, including 19 children with SMA1 and 8 with SMA2. The type of SMA was diagnosed by a neurologist prior to pharmacological treatment, on the basis of genetic testing and the functional status. In the study group, 13 children were unable to sit independently, 13 children were able to sit without support but did not walk, and one participant was able to walk. Single thoraco-lumbar or double scoliosis was present in 16 children. Hip subluxation was diagnosed in 19 participants.

The participants were divided into two groups, depending on the scale that was used to assess motor functions. The functional status of 15 children was assessed with the CHOP INTEND scale (CHOP group) and of 12 children with the HFMSE scale (HFMSE group). PO was measured in 14 participants who were able to sit up independently, and muscle strength was assessed in 13 children (Table 1).

Table 1 provides general information about the participants. The values of the postural parameters, ranges of motion, motor function score, and muscle strength measurements are presented in Table 2.

### 3.1. Comparison of Ranges of Motion, Muscle Strength and Postural Parameters in Groups

Rotation ranges in the cervical spine (CRS) (*p* = 0.413), LEE (*p* = 0.503), LHF (*p* = 0.882), and LHE (*p* = 0.479) in the hip joints did not differ between groups. In children examined with the CHOP scale, greater limitation of flexion in the shoulder joints was observed (*p* = 0.040) (Table 2).

A comparison of the force measurements showed higher values of NEL (*p* = 0.027), SFL (*p* = 0.015), SFR (*p* = 0.046), SBL (*p* = 0.005), SBR (*p* = 0.005), PFL (*p* = 0.022), PBL (*p* = 0.015), and PBR (*p* = 0.022) in the HFMS group. The sum of all parameters of the RS force also differed significantly (*p* = 0.007). Children from the CHOP group turned out to be significantly weaker than participants from the HFMSE group. There were no differences between the parameters of NFL (*p* = 0.281), NFR (*p* = 0.084), NER (*p* = 0.053), and PFR (*p* = 0.073) (Table 2).

The analysis showed no significant differences between the measurements of the Cobb angle (*p* = 0.913), ATRT (*p* = 0.214), ATRL (*p* = 0.561), SATRU (*p* = 0.747), and SATRL (*p* = 0.546) for the CHOP and HFMSE groups (Table 2). Due to the fact that, in the group assessed with the CHOP INTEND scale, only 2 children were able to sit independently, PO was not analyzed in this group, and this value was not compared between the groups.

### 3.2. Relationships between MF and Other Parameters in Groups

There were more significant correlations between the measurements in the CHOP group than in the HFMSE group (Figure 2 and Figure 3).

The MF of children examined with CHOP INTEND did not show a moderate or high relationship with the ROM in the cervical spine and limb joints. A low negative correlation was observed between the motor function and restriction of hip extension in this group (Figure 2). In the HFMSE group, smaller contractures of hamstrings were observed in children who obtained higher scores in the scale (Figure 3).

Both groups showed a high negative correlation between the MF and scoliosis, expressed in Cobb angle values (Figure 2 and Figure 3) (Table 2). Children with larger spinal deformities showed a worse functional status. Additionally, in the CHOP group, a negative relationship was observed between the MF score in the scale and the chest deformity (ATRT −0.573 *, *p* = 0.032; ATRL −0.697 **, and *p* = 0.006). As the deformation of the chest increased, the functional abilities of children deteriorated.

No relationship was found in any of the groups between the MF and the sum of RS muscle strength measurements (Figure 2 and Figure 3). However, a detailed analysis showed a significant relationship between the values of individual force measurements and the MF in both groups. In the CHOP group, a correlation was observed between the measurement of strength in the lower trunk and functional status, and in the HFMSE group, such a relationship was demonstrated by one of the measurements of strength in the upper trunk (Table 3).

### 3.3. Relationships between Spinal Deformity, ROM and Muscle Strength

The analysis of the data of all 27 participants showed that there was a significant correlation between the values of the Cobb angle and contractures in the shoulder joints, hip flexors and hamstrings (Table 4, Figure 4). However, a detailed analysis demonstrated that significant correlations between the Cobb angle values and ROM in the shoulder and hip joints occur only in the group of children assessed with the CHOP INTEND scale (Figure 2). In the HMFSE group, no correlation was found between the Cobb angle values and contractures in the limb joints and the ranges of neck rotation (Figure 3).

Additionally, in participants from the CHOP group, the ranges of neck rotation decreased with the increase of the chest deformity (ATRT −0.621, *p* = 0.018; SATRL −0.611, *p* = 0.015). In the HMFS group, on the other hand, there was a significant correlation between PO and CRS (−0.807, *p* = 0.002). As the CRS ranges increased, the PO value decreased, which means that children with a symmetrically positioned pelvis had greater ranges of neck rotation. At the same time, chest deformity influenced the range of motion in shoulder joints (0.591 *, *p* = 0.043).

Postural parameters were also associated with measurements of muscle strength. In the CHOP group, a correlation was observed between the size of the Cobb angle and the overall strength of the RS muscles (Figure 2). In stronger children, there were smaller curvatures of the spine. No such relationship was found in the HMFSE group (Figure 3). The analysis in the whole group did not confirm the significance of the relationship between the overall the muscle strength, RS, and the Cobb angle values (Table 4, Figure 4). On the other hand, relations between the RS, thoracic deformity, and oblique position of the pelvis in children able to sit independently were shown (Table 4).

The overall RS strength coefficient obtained in the group of 27 children with SMA was not correlated with the range of motion. However, associations were observed between the range of motion of the neck and shoulder joints and some force measurements. The increase in force was accompanied by greater ranges of motion (Table 4).

The results obtained in the CHOP and HFMSE groups differed from one another. In the HFMSE group, the ranges of head rotation showed a strong relationship with six parameters of muscle strength and the RS coefficient. Increases in the strength of the muscles, flexing and straightening the cervical spine, and the muscles of the upper and lower trunk were accompanied by an increase in the ranges of rotation in the cervical spine (Table 3). There was no such correlation in the group of children assessed with the CHOP INTEND scale (Table 3). On the other hand, in children assessed by this scale, a strong correlation was observed between the RS coefficient and contractures in the elbow joints. In addition, an increase in the contractures in the elbow joints showed a relationship with an increase in the strength of the neck flexing muscles. In the HFMS group, four strong negative correlations were observed between the measurements of strength in the upper and lower trunk and contractures in the elbow joints (Table 3). In both groups, there were seven significant positive correlations between individual trunk muscle strength measurements and the range of motion in the lower limb joints (Table 3).

### 3.4. Relationships between Ranges of Motion

Participants showed significant relationships between the ROM in the neck and joints of the limbs, with more significant relationships in the CHOP group. In this group, a significant negative correlation between CRS and limitations of the ROM in the shoulder joints and elbows was found. The intensification of contractures in the joints of the upper limbs was accompanied by a decrease in the rotational ranges of the neck. Additionally, there were significant relationships between the contractures in the upper and lower limbs. In the HFMS group, significant relationships were found between the contractures in the joints of the upper and lower limbs (Figure 3).

Figure 4 shows moderate and high significant correlations between the assessed parameters in the study group.

## 4. Discussion

Many positive changes in the treatment of SMA patients, due to the implementation of pharmacological treatment, have been seen in recent years. New therapeutic options make it possible for many patients to improve their functional status or slow down the progression of the disease, but, on the other hand, they are a significant financial burden under the reimbursed care system. The treatment process should, therefore, be conducted in an optimal way, ensuring a maximum effect with the financial costs down to a minimum. In order to optimize therapeutic procedures, it is important to systematically assess the structures and functions of the body and the level of activity, in accordance with the International Classification of Functioning, Disability and Health. Knowledge about the impact of particular structures and functions of the body on motor skills is extremely valuable.

In this study, a detailed analysis of the MF, as well as the selected structures and functions of the bodies of children with SMA, was carried out using functional scales and tests with proven reliability. Weaker children were assessed with the CHOP INTEND scale and stronger children, despite the earlier diagnosis of SMA1, were assessed with the HFMSE scale.

The relationships between MF and factors such as muscle strength, range of motion, and spine and chest deformities were analyzed. It has been shown that the motor skills of children with SMA can be affected by the strength of certain groups of muscles in the trunk (upper trunk in the HFMSE group and lower trunk in the CHOP group) and contractures in the joints of the lower limbs. A strong association of the MF with spine, thoracic, and pelvic deformities was observed in both groups.

In the past, the relationship between the MF and the strength of the muscles in individuals with SMA has been pointed out by other researchers. Kroksmark et al. showed relationships between the strength of selected limb muscles and the way people with SMA2 and SMA3 perform certain movements. Motor functions, in the form of 20 movements, have been recorded and scored. Additionally, muscle strength was measured. It was shown that decreased muscle strength caused a deterioration of the motor skills in the study group. The authors also observed that proximal weakness was greater than distal, and the upper limbs were stronger than the lower ones [2]. Merlini et al. investigated the correlation between the values of limb strength and the gait time at a distance of 10 m (standing up from the floor and climbing stairs), showing significant relationships [4]. In participants with SMA2 and SMA3, Kauffman et al. observed a significant relationship between HFMSE and elbow flexion, knee extension, and knee flexion strength [6]. Febrer et al. reported differences in muscle strength in walking and non-walking patients [5]. Relationships between the strength of selected muscle groups in the upper limbs and motor skills assessed by functional scales have also been discussed. 

In previous studies by Stępień et al., it was shown that the strength of the neck and trunk muscles in children with SMA1 and SMA2 (non-sitters and sitters), that was not treated pharmacologically, was significantly different from the strength of children with SMA 3 (sitters and walkers). However, the authors did not use functional scales, assessing only the possibility of rotation from lying on the back to lying on the side [28]. This study is the first to analyze the relationship between motor skills, assessed by reliable scales, and muscle strength in the neck and trunk. We found that, in the group of weaker children assessed by the CHOP INTEND scale, MF depended on the strength of the lower trunk muscles and in the HFMSE group, it was related to the strength of the upper trunk muscles. This information encourages an analysis of the effect of trunk strengthening exercises on motor function in children with SMA.

In the past, it has been shown that movement abilities may depend on the ROM in joints. In participants with SMA2 and SMA3, Salazar et al. observed that slight ROM limitation in the hip and knee joints may affect motor abilities [30]. The authors assessed ranges of motion in the joints of the lower limbs and motor functions using the HFMSE scale. The authors suggested that contractures should be included in the assessment of the motor functions. In our study, in children assessed with the HFMSE scale, a significant relationship was demonstrated between the MF scale scores and LHF ranges. More fit children showed smaller hamstrings contractures. The results of our study, thus, support Salazar’s concept that the limitation of the range of motion in the joints of the lower limbs affects the motor skills of patients with SMA and should be considered as part of the assessment of motor function.

The results of this study show a strong relationship between MF and the size of the scoliosis, severity of chest deformity, and oblique position of the pelvis. In both the CHOP and HFMSE study groups, children with larger deformations showed worse results when assessed by functional scales. This means that the deformation of the spine, chest, and pelvis, which increases with age, may limit the functioning of the child.

Scoliosis affected not only the functional status of the examined children, but also the ROM. In the CHOP group, contractures in the shoulder and hip joints increased with the increase of scoliosis. It is worth noting that limiting the range of extension or flexion in the hip joints accompanying scoliosis may make it difficult to stay in a standing or sitting position. In a study conducted by Stępień et al., in a group of patients with SMA, it was observed that children with scoliosis showed greater differences between the extension ranges in the hip joints than in children without scoliosis [10]. The limitation of mobility and asymmetry of movements in the hip joints, occurring in people with scoliosis, may contribute to dislocations in the hip joints, often described in the literature [31]. Therefore, it is advisable to regularly test the mobility of the hip joints in children with SMA, especially in those with scoliosis.

The development of spinal deformity leads to changes in other structures. We found that that chest deformities in the CHOP group and pelvic position in the HFMSE group were associated with ranges of neck rotation. We also found a relationship between the Cobb angle and PO values in the HFMSE group.

The impact of scoliosis on the ranges of neck rotation and other postural parameters in 74 children with SMA1, SMA2, and SMA3 (not treated pharmacologically) was discussed in an earlier study by Stępień et al. The authors, in addition to the limitations of the range of cervical rotation, showed that the ranges of neck rotation, chest shape, and pelvic alignment are interrelated and depend on the prevalence of scoliosis, age, and the type of SMA. Children with SMA1 and SMA2 showed more severe deformities and movement limitations, compared to the SMA3 and control groups [8]. Fujak et al., when analyzing the natural development of scoliosis, observed that the oblique position of the pelvis is common in children with SMA, including the youngest ones, and it is more severe in the SMA2 group, compared to the SMA3 type [7]. 

Spine deformity, and the accompanying changes in the structure of the chest and the position of the pelvis, in our study was also correlated with the measurements of the strength of the muscles of the neck and trunk. Significant correlations between muscle strength, the deformation of the chest, and the oblique position of the pelvis have been demonstrated in children able to sit independently. Children with larger deformities showed lower muscle strength values. It is difficult to relate these results to other publications because, to the best of our knowledge, no similar studies have been conducted so far. Due to the presented dependencies in the musculoskeletal system, the examination of a child with SMA should be detailed and include various parts of the body.

The above results indicate that the development of scoliosis in SMA patients adversely affects the MF, ROM, and muscle strength values and may influence quality of life. The negative impact of neuromuscular scoliosis on daily functioning has been noticed in the literature. It has been observed that deformation of the spine hinders daily care, functioning in a sitting and standing position, and may cause pain and respiratory disorders [32,33]. Progressive scoliosis is diagnosed in most children with SMA1 and SMA2, less often in children with SMA3 type [7,34]. Daily practice shows that scoliosis often makes it difficult for these children to properly control the head, maintain position, and make limb movements. In the population of individuals with SMA, there are additional pain symptoms, located in the cervical region, back, or limbs [35], as well as swallowing disorders [36]. The biomechanical relationships, described by us, between the shape of the chest, the position of the pelvis, and the ranges of rotation in the cervical region may indicate that both pain and swallowing disorders may be related to the deformation of the spine. The characteristic scoliotic three-plane loss of alignment of the spine and pelvis change the position of the articular surfaces, which may lead to overloads and microinjuries and require appropriate diagnostics [37]. In addition, immobilization in the form of traction systems, collars, or braces, often used in SMA patients with scoliosis, may lead to secondary muscle weakness [38]. However, these are hypotheses that require confirmation in future scientific projects.

Due to numerous disorders associated with deformation of the spine, preventing the development of scoliosis and the accompanying changes in the musculoskeletal system should be considered one of the main goals of treatment of children with SMA.

Relationships between ranges of motion and muscle strength measurements were observed in the participants in our study. An increase in muscle strength was accompanied by greater ranges of motion. This points to the need for treatment programs to include exercises to strengthen the muscles of the neck and trunk, to use stretching exercises, and to regularly evaluate these functions.

There are reports in the literature on the limitations of the ranges of motion in people with SMA [9,10]. However, the relationship between range of motion measurements made in different parts of the body has not yet been investigated. In our study, significant correlations were found between the ranges of motion within the neck, as well as the upper and lower limbs. Tension in one part of the body was accompanied by a limitation of motion in other parts of the body. More significant relationships were seen in the CHOP group, in which the values of force measurements were lower than in the HFMSE group. These results show the need for regular measurements of ranges of motion and the use of stretching exercises in various regions of the body, especially in weaker children with SMA and comorbid scoliosis.

The results of this study are of practical use and should be taken into account when assessing the effectiveness of treating patients with SMA. The deterioration or lack of improvement in the functional status of a child does not have to result from the ineffectiveness of the pharmacological treatment used, but it may result from the progression of scoliosis, intensification of chest deformities, or increasing contractures. So far, when assessing the effectiveness of treatment, these factors have not been taken into account, treating motor skills as the most important ones.

The results of this study confirm the validity of the guidelines for the care of patients with SMA, which include stretching, using strengthening exercises, training of daily activities, stimulating breathing, and appropriate positioning [14]. It is important to systematically assess the musculoskeletal system and to implement early prevention aimed at preventing scoliosis and contractures, as well as training to strengthen the muscles.

There are some limitations of the study; among them is the small size of the study group, and the large age diversity of the participants. Correlations between the MF, ROM, and muscle strength were analyzed in two small groups. This may affect the results obtained. In the future, it is worth conducting studies in larger and specific age groups. Another limitation of the study may be that the participants were treated with various medications that were not fully described.

## 5. Conclusions

The motor skills of children with SMA depend on various factors, such as range of motion and muscle strength, as well as deformities of the spine, chest and pelvis. These relationships are more pronounced in weaker children. The motor assessment of a child with SMA should, therefore, not be limited only to the use of motor scales, but also take into account postural parameters, as well as measurements of the range of motion and muscle strength. These factors should be considered when deciding whether to continue using medications.

The development of scoliosis adversely affects the motor function, range of motion, and muscle strength. The limitations of the range of motion and reduced muscle strength associated with scoliosis may hinder the daily functioning of the child. The musculoskeletal system should be systematically monitored, and the development of spine deformation should be prevented.

Movement ranges are related to motor skills and strength values. Regular measurements of ranges of motion are necessary, especially in weaker children with SMA and comorbid scoliosis.

Accurate functional tests are the basis of therapeutic procedures. The analysis of the relationships between postural parameters, range of motion, and muscle strength brings a lot of valuable information that is useful for planning therapy.

## Figures and Tables

**Figure 1 ijerph-18-09134-f001:**
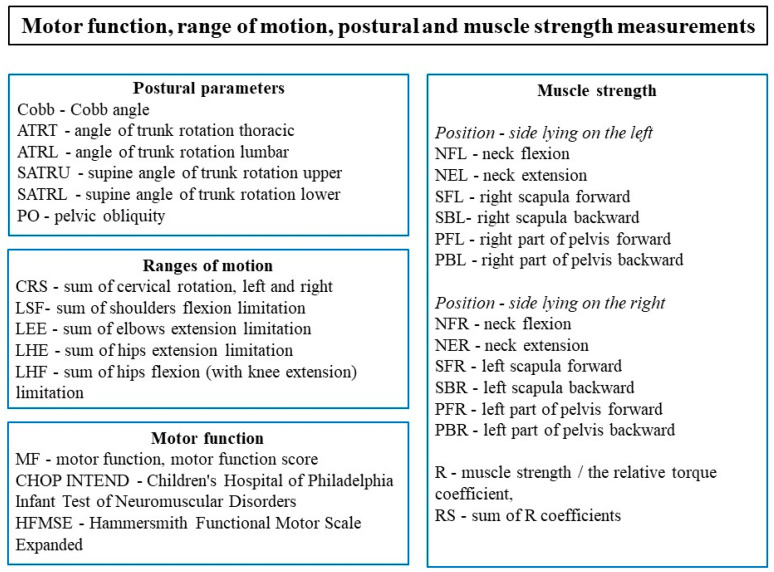
Motor function, range of motion, postural, and muscle strength measurements with abbreviations.

**Figure 2 ijerph-18-09134-f002:**
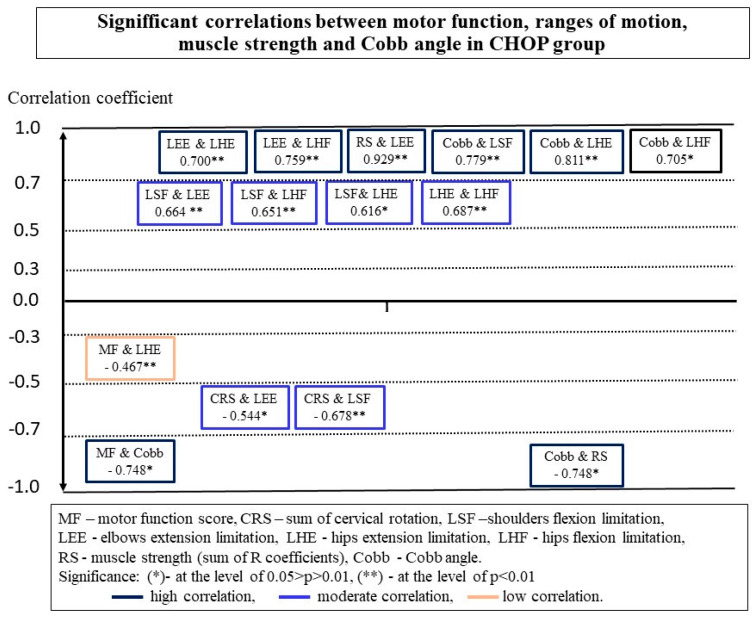
Significant correlations between motor function, range of motion, muscle strength, and Cobb angle in the CHOP group.

**Figure 3 ijerph-18-09134-f003:**
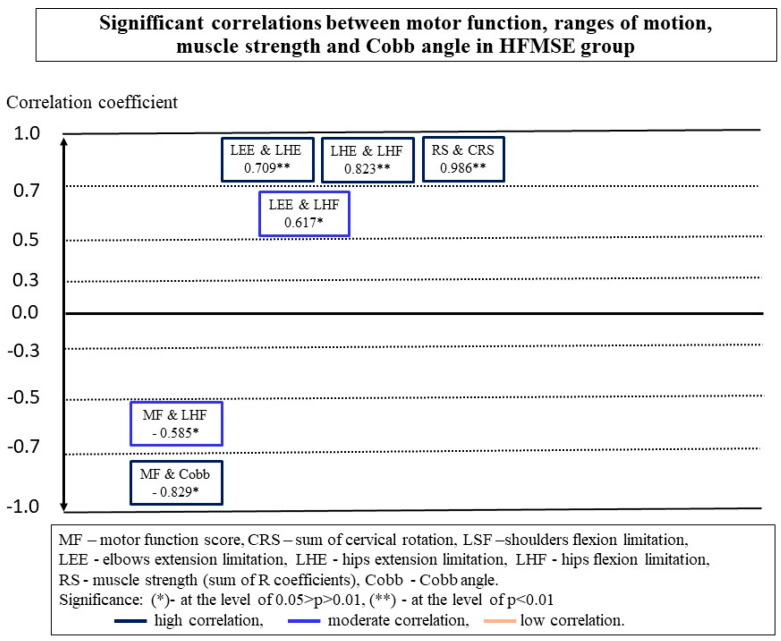
Significant correlations between motor function, range of motion, muscle strength, and Cobb angle in the HFMSE group.

**Figure 4 ijerph-18-09134-f004:**
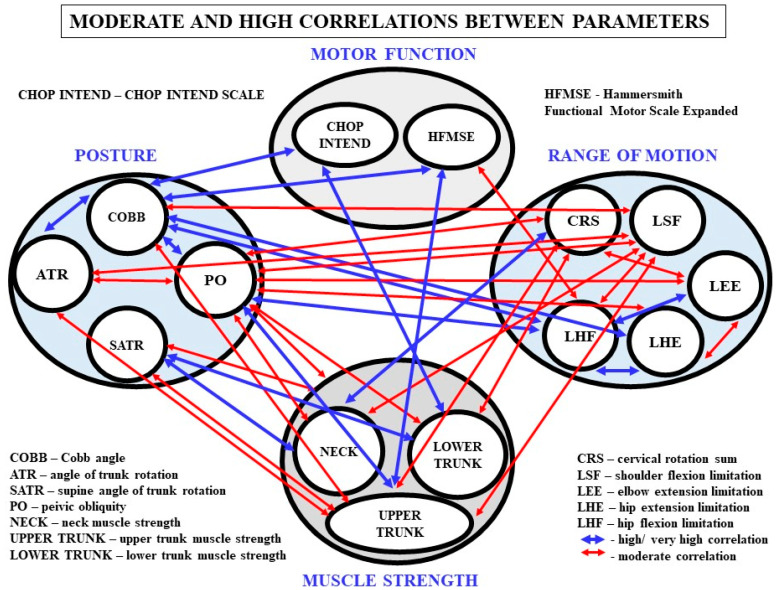
Graphical interpretation of moderate, high, and very high correlations between the examined parameters; a total of 14 high/very high correlations and 22 moderate correlations between motor function, muscle strength, range of motion, and postural parameters were observed in the study group.

**Table 1 ijerph-18-09134-t001:** Characteristics of participants.

Mean Values (±SD) Min-Max	All Participants (n = 27)	CHOP Group (n = 15)	HFMSE Group (n = 12)
Age [years]	5.23 ± 3.49	4.66 ± 3.39	5.95 ± 3.62 *
Min–Max	0.80–14.40	0.80–13.50	2.50–14.40
Body mass [kg]	15.64 ± 8.73	12.71 ± 5.72	19.42 ± 10.47 **
Min-Max	6.00–37.00	6.00–27.00	10.00–37.00
Body height [m]	1.04 ± 0.22	0.99 ± 0.23	1.10 ± 0.20 *
Min-Max	0.65–1.50	0.65–1.50	0.80–1.40
Gender			
Girls	11 [40.7%]	7 [46.7%]	4 [33.3%]
Boys	16 [59.3%]	8 [53.3%]	8 [66.7%]
Type of SMA [n]			
SMA1	19 [70.4%]	15 [100.0%]	4 [33.3%]
SMA2	8 [29.6%]	-	8 [66.7%]
Motor abilities [n]			
Non-sitters	13 [ 48.1%]	13 [86.7%]	-
Sitters	13 [48.1%]	2 [13.3%]	11 [91.7%]
Walkers	1 [3.6%]	-	1 [8.3%]
SMN2 copy number [n]			
2 copies	13 [48.1%]	12 [80.0%]	1 [8.3%]
3 copies	12 [44.4%]	3 [20.0%]	9 [75.0%]
4 copies	2 [7.5%]	-	2 [16.7%]
Hip subluxation	19 [70.4%]	12 [80.0%]	7 [58.3%]
Scoliosis [n]	16 [59.3%]	10 [66.7%]	6 [50.0%]
- single scoliosis	10 [37.0%]	6 [40.0%]	4 [33.3%]
- double scoliosis	6 [22.3%]	4 [26.7%]	2 [16.7%]

Note: n—number of participants, SMN2–SMN2 gene. Significance of difference between CHOP and HFMSE groups: (*)—at the level of 0.01 < *p* < 0.05, (**)—at the level of 0.001 ≤ *p* ≤ 0.01.

**Table 2 ijerph-18-09134-t002:** Values of the postural parameters, ranges of motion, motor function score, and muscle strength measurements.

Mean Values (±SD) Min-Max	All Participants n = 27	CHOP Group n = 15	HFMSE Group n = 12
Postural parameters			
Cobb angle [°]	38.20 ± 25.40 (n = 16)	39.10 ± 28.20 (n = 10)	36.83 ± 22.35 (n = 6)
Min-Max	14.00–90.00	17.00–90.00	14.00–78.00
ATRT [°]	7.96 ± 9.37	10.29 ± 11.61	5.25 ± 5.05
Min-Max	0.00–40.00	0.00–40.00	0.00–13.00
ATRL [°]	3.19 ± 3.55	2.43 ± 2.34	4.08 ± 4.54
Min-Max	0.00–15.00	0.00–5.00	0.00–15.00
SATRU [°]	3.48 ± 3.25	3.67 ± 3.37	3.25 ± 3.22
Min-Max	0.00–11.00	0.00–10.00	0.00–11.00
SATRL [°]	2.81 ± 3.01	3.33 ± 3.64	2.17 ± 1.95
Min-Max	0.00–11.00	0.00–11.00	0.00–5.00
PO [°]	4.71 ± 3.58 (n = 14)	1.00 ± 1.41 (n = 2)	5.33 ± 3.47 (n = 12)
Min-Max	0.00–12.00	0.00–2.00	1.00–12.00
Ranges of motion			
CRS [°] Min-Max	60.89 ± 15.72 14.00–80.00	58.07 ± 18.22 14.00–80.00	64.42 ± 11.72 36.00–80.00
LSF [°] Min-Max	34.15 ± 57.48 0.00–170.00	57.20 ± 69.14 0.00–170.00	5.33 ± 9.70* 0.00–24.00
LEE [°] Min-Max	14.78 ± 25.08 0.00–80.00	19.80 ± 30.06 0.00–80.00	8.50 ± 6.12 0.00–54.00
LHE [°]	44.89 ± 44.98	44.60 ± 48.45	45.25 ± 42.36
Min-Max	0.00–129.00	0.00–129.00	0.00–92.00
LHF [°]	66.89 ± 45.45	72.13 ± 45.57	60.33 ± 46.43
Min-Max	0.00–146.00	0.00–146.00	0.00–134.00
Motor function			
MF score	-	28.27 ± 14.04	21.67 ± 15.45
Min-Max	-	2.00–49.00	4.00–54.00
Muscle strength [Nm/kg]			
Neck			
	n = 13	n = 7	n = 6
NFL R & NFR R	0.18 ± 0.06 & 0.18 ± 0.07	0.16 ± 0.05 & 0.15 ± 0.06	0.21 ± 0.08 * & 0.22 ± 0.08
Min	0.09 & 0.10	0.09 & 0.10	0.13 & 0.13
Max	0.34 & 0.33	0.22 & 0.27	0.34 & 0.33
NEL R & NER R	0.31 ± 0.12 & 0.32 ± 0.15	0.24 ± 0.09 & 0.23 ± 0.13	0.39 ± 0.11& 0.41 ± 0.12
Min	0.12 & 0.10	0.12 & 0.10	0.24 & 0.26
Max	0.58 & 0.58	0.39 & 0.45	0.58 & 0.58
Scapula–upper trunk			
SFL R & SFR R	0.20 ± 0.09 & 0.25 ± 0.12	0.15 ± 0.07 & 0.18 ± 0.10	0.27 ± 0.07 * & 0.32 ± 0.09 *
Min	0.07 & 0.07	0.07 & 0.07	0.19 & 0.19
Max	0.38 & 0.43	0.29 & 0.35	0.38 & 0.43
SBL R & SBR R	0.34 ± 0.20 & 0.28 ± 0.15	0.22 ± 0.07 & 0.20 ± 0.05	0.49 ± 0.21 ** & 0.37 ± 0.18 **
Min	0.12 & 0.11	0.12 & 0.11	0.30 & 0.24
Max	0.89 & 0.71	0.32 & 0.27	0.89 & 0.71
Pelvis–lower trunk			
PFL R & PFR R	0.26 ± 0.12 & 0.28 ± 0.11	0.19 ± 0.07 & 0.23 ± 0.10	0.34 ± 0.13 * & 0.35 ± 0.08
Min	0.08 & 0.13	0.08 & 0.13	0.21 & 0.27
Max	0.55 & 0.48	0.29 & 0.37	0.55 & 0.48
PBL R & PBR R	0.54 ± 0.22 & 0.52 ± 0.16	0.41 ± 0.21 & 0.44 ± 0.14	0.70 ± 0.09 * & 0.62 ± 0.14 *
Min	0.12 & 0.25	0.12 & 0.25	0.55 & 0.42
Max	0.78 & 0.75	0.67 & 0.62	0.78 & 0.75
Muscle strength sum			
RS	3.68 ± 1.29	2.80 ± 0.66	4.70 ± 1.06 **
Min-Max	1.64–6.17	1.64–3.57	3.50–6.17

Note: n—number of participants, ATRT—angle of trunk rotation thoracic, ATRL—angle of trunk rotation lumbar, SATRU—supine angle of trunk rotation upper, SATRL—supine angle of trunk rotation lower, PO—pelvic obliquity, CRS—sum of cervical rotation to the left and right, LSF—sum of shoulder flexion limitation, LEE—sum of elbow extension limitation, LHE—sum of hip extension limitation, LHF—sum of hip flexion with knee extension limitation in both hip joints, MF—motor function, R—muscle strength / the relative torque coefficient, NFL—neck flexion in side lying on the left, NEL—neck extension in side lying on the left, SFL—right scapula forward in side lying on the left, SBL—right scapula backward in side lying on the left, PFL—right part of pelvis forward in side lying on the left, PBL—right part of pelvis backward in side lying on the left, NFR—neck flexion in side lying on the right, NER—neck extension in side lying on the right, SFR—left scapula forward in side lying on the right, SBR—left scapula backward in side lying on the right, PFR—left part of pelvis forward in side lying on the right, PBR—left part of pelvis backward in side lying on the right, RS—sum of R coefficients, N—newton, m—meter, kg—kilogram, and MF—motor function score. Significance of difference between CHOP and HFMSE groups: (*)—at the level of 0.01 < *p* < 0.05, (**)—at the level of 0.001 ≤ *p* ≤ 0.01.

**Table 3 ijerph-18-09134-t003:** Relationships between individual values of muscle strength, MF score, and ROM in the CHOP and HFMS groups.

**Chop Intend (n = 7)**						
**Muscle Strength**	**MF**	**CRS**	**LSF**	**LEE**	**LHE**	**LHF**
NFL (Nm/kg)	−0.071 *p* = 0.879	0.071 *p* = 0.879	−0.214 *p* = 0.645	−0.036 *p* = 0.939	0.108 *p* = 0.818	0.643 *p* = 0.119
NEL (Nm/kg)	0.571 *p* = 0.180	−0.036 *p* = 0.939	−0.214 *p* = 0.645	0.643 *p* = 0.119	0.180 *p* = 0.699	0.786 * *p* = 0.036
SFL (Nm/kg)	0.360 *p* = 0.427	−0.505 *p* = 0.248	0.108 *p* = 0.818	0.649 *p* = 0.115	0.555 *p* = 0.196	0.829 * *p* = 0.021
SBL (Nm/kg)	0.750 *p* = 0.052	0.000 *p* = 1.000	−0.250 *p* = 0.589	0.286 *p* = 0.535	−0.018 *p* = 0.969	−0.286 *p* = 0.535
PFL (Nm/kg	0.090 *p* = 0.848	−0.324 *p* = 0.478	−0.054 *p* = 0.908	0.180 *p* = 0.699	0.900 ** *p* = 0.006	0.234 *p* = 0.613
PBL (Nm/kg)	0.750 *p* = 0.052	−0.357 *p* = 0.432	−0.036 *p* = 0.939	0.500 *p* = 0.253	0.216 *p* = 0.641	−0.143 *p* = 0.760
NFR (Nm/kg)	0.721 *p* = 0.068	−0.144 *p* = 0.758	−0.072 *p* = 0.878	0.829 * *p* = 0.021	−0.109 *p* = 0.816	0.559 *p* = 0.192
NER (Nm/kg)	−0.071 *p* = 0.879	0.464 *p* = 0.294	−0.107 *p* = 0.819	0.393 *p* = 0.383	0.180 *p* = 0.690	0.536 *p* = 0.215
SFR (Nm/kg)	0.357 *p* = 0.432	−0.571 *p* = 0.180	0.393 *p* = 0.383	0.393 *p* = 0.383	0.306 *p* = 0.504	−0.214 *p* = 0.645
SBR (Nm/kg)	−0.107 *p* = 0.819	0.071 *p* = 0.879	0.071 *p* = 0.879	0.500 *p* = 0.253	0.378 *p* = 0.403	0.714 *p* = 0.071
PFR (Nm/kg)	0.144 *p* = 0.758	−0.180 *p* = 0.699	0.360 *p* = 0.427	0.721 *p* = 0.068	0.700 *p* = 0.008	0.541 *p* = 0.210
PBR (Nm/kg)	0.821 * *p* = 0.023	0.000 *p* = 1.000	−0.143 *p* = 0.760	0.714 *p* = 0.071	−0.018 *p* = 0.969	0.250 *p* = 0.589
**Hammersmith Functional Motor Scale Expanded (n = 6)**						
**Muscle strength**	**MF**	**CRS**	**LSF**	**LEE**	**LHE**	**LHF**
NFL (Nm/kg)	0.580 *p* = 0.228	0.870 * *p* = 0.024	0.000 *p* = 1.000	−0.515 *p* = 0.296	0.261 *p* = 0.618	0.116 *p* = 0.827
NEL (Nm/kg)	0.600 *p* = 0.208	0.943 ** *p* = 0.005	0.135 *p* = 0.798	−0.754 *p* = 0.084	−0.143 *p* = 0.787	−0.029 *p* = 0.957
SFL (Nm/kg)	0.714 *p* = 0.111	−0.139 *p* = 0.666	0.270 *p* = 0.604	−0.232 *p* = 0.658	0.086 *p* = 0.872	0.086 *p* = 0.872
SBL (Nm/kg)	0.829 * *p* = 0.042	0.943 ** *p* = 0.005	−0.270 *p* = 0.604	−0.725 *p* = 0.103	−0.314 *p* = 0.544	−0.429 *p* = 0.397
PFL (Nm/kg)	0.486 *p* = 0.329	0.600 *p* = 0.208	0.507 *p* = 0.305	0.116 *p* = 0.827	0.600 *p* = 0.208	0.371 *p* = 0.468
PBL (Nm/kg)	0.314 *p* = 0.544	0.771 *p* = 0.072	−0.101 *p* = 0.848	−0.928 ** *p* = 0.008	−0.257 *p* = 0.623	0.029 *p* = 0.957
NFR (Nm/kg)	0.522 *p* = 0.288	0.696 *p* = 0.125	0.274 *p* = 0.599	−0.088 *p* = 0.868	0.580 *p* = 0.228	0.290 *p* = 0.577
NER (Nm/kg)	0.486 *p* = 0.329	0.829 * *p* = 0.042	0.372 *p* = 0.468	−0.580 *p* = 0.228	−0.086 *p* = 0.872	0.143 *p* = 0.787
SFR (Nm/kg)	0.200 *p* = 0.704	0.771 *p* = 0.072	0.304 *p* = 0.558	−0.812 * *p* = 0.050	0.086 *p* = 0.872	0.314 *p* = 0.544
SBR (Nm/kg)	0.714 *p* = 0.111	0.829 * *p* = 0.042	−0.507 *p* = 0.305	−0.841 * *p* = 0.036	−0.486 *p* = 0.329	−0.543 *p* = 0.266
PFR (Nm/kg)	−0.257 *p* = 0.623	0.086 *p* = 0.872	0.372 *p* = 0.468	−0.029 *p* = 0.957	0.829 * *p* = 0.042	0.543 *p* = 0.266
PBR (Nm/kg)	0.429 *p* = 0.397	0.886 * *p* = 0.019	0.135 *p* = 0.798	−0.812 * *p* = 0.050	−0.086 *p* = 0.872	0.143 *p* = 0.787

Note: n—number of participants, CRS-sum of cervical rotation to the left and right, LSF-sum of shoulders flexion limitation, LEE-sum of elbows extension limitation, LHE-sum of hips extension limitation, LHF-sum of hips flexion with knee extension limitation in both hip joints, NFL-neck flexion in side lying on the left, NEL-neck extension in side lying on the left, SFL-right scapula forward in side lying on the left, SBL-right scapula backward in side lying on the left, PFL-right part of pelvis forward in side lying on the left, PBL-right part of pelvis backward in side lying on the left, NFR-neck flexion in side lying on the right, NER-neck extension in side lying on the right, SFR-left scapula forward in side lying on the right, SBR-left scapula backward in side lying on the right, PFR-left part of pelvis forward in side lying on the right, PBR-left part of pelvis backward in side lying on the right, and N—newton, m—meter, kg—kilogram. Significance: (*)—at the level of 0.01 < *p* < 0.05, (**)—at the level of 0.001 ≤ *p* ≤ 0.01.

**Table 4 ijerph-18-09134-t004:** Relationships between spinal deformity, ROM and muscle strength in all participants.

	Cobb(^o^) n = 16	ATRT(^o^) n = 27	ATRL(^o^) n = 27	SATRU(^o^) n = 27	SATRL(^o^) n = 27	PO(^o^) n = 14	CRS(^o^) n = 27	LSF(^o^) n== 27	LEE(^o^) n = 27	LHE(^o^) n = 27	LHF(^o^) n = 27
ATRT (^o^) n = 27	0.618 * *p* = 0.011	NA	-	-	-	-	-	-	-	-	-
ATRL (^o^) n = 27	0.708 ** *p* = 0.002	0.501 ** *p* = 0.009	NA	-	-	-	-	-	-	-	-
SATRU (^o^) n = 27	0.021 *p* = 0.940	0.257 *p* = 0.205	0.254 *p* = 0.210	NA	-	-	-	-	-	-	-
SATRL (^o^) n = 27	0.353 *p* = 0.180	0.469 * *p* = 0.016	0.165 *p* = 0.420	0.078 *p* = 0.700	NA	-	-	-	-	-	-
PO (^o^) n = 14	0.740 ** *p* = 0.001	0.537 ** *p* = 0.005	0.457 * *p* = 0.019	0.308 *p* = 0.126	0.155 *p* = 0.499	NA	-	-	-	-	-
CRS (^o^) n = 27	−0.410 *p* = 0.115	−0.373 *p* = 0.061	−0.224 *p* = 0.272	−0.301 *p* = 0.128	−0.358 *p* = 0.067	−0.649 ** *p* = 0.000	NA	-	-	-	-
LSF (^o^) n = 27	0.607 * *p* = 0.013	0.649 ** *p* = 0.000	0.337 *p* = 0.092	0.337 *p* = 0.092	0.418 * *p* = 0.030	0.646 ** *p* = 0.000	−0.404 * *p* = 0.037	NA	-	-	-
LEE (^o^) n = 27	0.483 *p* = 0.058	0.287 *p* = 0.155	0.217 *p* = 0.286	0.082 *p* = 0.683	0.417 * *p* = 0.030	0.612 ** *p* = 0.001	−0.554 ** *p* = 0.003	0.479 * *p* = 0.011	NA	-	-
LHE (^o^) n = 27	0.816 ** *p* = 0.001	0.241 *p* = 0.235	0.318 *p* = 0.113	−0.081 *p* = 0.686	0.322 *p* = 0.101	0.665 ** *p* = 0.001	−0.353 *p* = 0.070	0.428 * *p* = 0.026	0.675 ** *p* = 0.001	NA	-
LHF (^o^) n = 27	0.718 ** *p* = 0.002	0.498 ** *p* = 0.010	0.497 ** *p* = 0.010	−0.034 *p* = 0.868	0.253 *p* = 0.202	0.717 ** *p* = 0.001	−0.401 * *p* = 0.038	0.509 ** *p* = 0.007	0.709 ** *p* = 0.001	0.742 ** *p* = 0.001	NA
NFL (Nm/kg) n = 13	−0.352 *p* = 0.261	−0.295 *p* = 0.328	0.182 *p* = 0.551	−0.370 *p* = 0.213	−0.292 *p* = 0.333	−0.266 *p* = 0.380	0.543 *p* = 0.055	−0.396 *p* = 0.181	−0.292 *p* = 0.333	0.076 *p* = 0.804	0.202 *p* = 0.507
NEL (Nm/kg) n = 13	−0.520 *p* = 0.083	−0.412 *p* = 0.162	−0.072 *p* = 0.816	−0.603 * *p* = 0.029	−0.166 *p* = 0.587	−0.664 * *p* = 0.013	0.658 * *p* = 0.014	−0.617 * *p* = 0.025	−0.172 *p* = 0.575	−0.127 *p* = 0.680	0.019 *p* = 0.950
SFL (Nm/kg) n = 13	−0.069 *p* = 0.831	−0.282 *p* = 0.350	0.208 *p* = 0.495	−0.446 *p* = 0.126	0.150 *p* = 0.626	−0.321 *p* = 0.285	0.306 *p* = 0.310	−0.430 *p* = 0.142	−0.058 *p* = 0.850	0.164 *p* = 0.592	0.114 *p* = 0.710
SBL (Nm/kg) n = 13	−0.589 * *p* = 0.044	−0.564 * *p* = 0.045	−0.136 *p* = 0.657	−0.502 *p* = 0.081	0.013 *p* = 0.967	−0.842 ** *p* = 0.001	0.588 * *p* = 0.034	−0.659 * *p* = 0.014	−0.349 *p* = 0.243	−0.239 *p* = 0.432	−0.516 *p* = 0.071
PFL (Nm/kg) n = 13	0.172 *p* = 0.594	−0.232 *p* = 0.445	0.478 *p* = 0.099	−0.330 *p* = 0.271	0.346 *p* = 0.247	−0.355 *p* = 0.233	0.430 *p* = 0.143	−0.342 *p* = 0.252	−0.111 *p* = 0.719	0.415 *p* = 0.158	0.012 *p* = 0.968
PBL (Nm/kg) n = 13	−0.442 *p* = 0.151	−0.221 *p* = 0.468	−0.247 *p* = 0.416	−0.505 *p* = 0.078	0.296 *p* = 0.326	−0.638 * *p* = 0.019	0.380 *p* = 0.201	−0.492 *p* = 0.088	−0.276 *p* = 0.361	−0.182 *p* = 0.552	−0.350 *p* = 0.241
NFR (Nm/kg) n = 13	−0.349 *p* = 0.261	−0.398 *p* = 0.178	0.054 *p* = 0.862	−0.723 ** *p* = 0.005	0.091 *p* = 0.768	−0.524 *p* = 0.066	0.492 *p* = 0.088	−0.467 *p* = 0.108	0.192 *p* = 0.530	−0.062 *p* = 0.839	0.207 *p* = 0.498
NER (Nm/kg) n = 13	−0.257 *p* = 0.421	−0.188 *p* = 0.538	0.055 *p* = 0.859	−0.250 *p* = 0.410	0.003 *p* = 0.993	−0.598 * *p* = 0.031	0.769 ** *p* = 0.002	−0.549 *p* = 0.052	−0.158 *p* = 0.607	−0.086 *p* = 0.781	0.074 *p* = 0.809
SFR (Nm/kg) n = 13	−0.226 *p* = 0.480	−0.055 *p* = 0.858	−0.006 *p* = 0.985	−0.599 * *p* = 0.031	0.442 *p* = 0.130	−0.445 *p* = 0.128	0.327 *p* = 0.275	−0.164 *p* = 0.593	−0.199 *p* = 0.515	−0.008 *p* = 0.979	−0.179 *p* = 0.558
SBR (Nm/kg) n = 13	−0.274 *p* = 0.388	−0.279 *p* = 0.355	0.086 *p* = 0.781	−0.148 *p* = 0.630	−0.020 *p* = 0.949	−0.613 * *p* = 0.026	0.534 *p* = 0.060	−0.662 * *p* = 0.014	−0.321 *p* = 0.285	−0.080 *p* = 0.795	−0.228 *p* = 0.455
PFR (Nm/kg) n = 13	0.294 *p* = 0.353	0.044 *p* = 0.886	0.339 *p* = 0.257	−0.498 *p* = 0.083	0.527 *p* = 0.064	−0.325 *p* = 0.278	0.276 *p* = 0.362	−0.159 *p* = 0.605	0.080 *p* = 0.794	0.464 *p* = 0.110	0.115 *p* = 0.709
PBR (Nm/kg) n = 13	−0.505 *p* = 0.094	−0.331 *p* = 0.269	−0.168 *p* = 0.582	−0.701 ** *p* = 0.008	0.169 *p* = 0.581	−0.685 ** *p* = 0.010	0.605 * *p* = 0.028	−0.537 *p* = 0.058	−0.122 *p* = 0.692	−0.198 *p* = 0.516	−0.063 *p* = 0.837
RS (Nm/kg) n = 13	−0.329 *p* = 0.296	−0.335 *p* = 0.264	0.022 *p* = 0.942	−0.588 * *p* = 0.034	0.206 *p* = 0.500	−0.643 * *p =* 0.018	0.537 *p* = 0.058	−0.549 *p* = 0.052	−0.113 *p* = 0.712	−0.055 *p* = 0.858	−0.084 *p* = 0.785

Note: n—number of participants, ATRT—angle of trunk rotation thoracic, ATRL—angle of trunk rotation lumbar, SATRU—supine angle of trunk rotation upper, SATRL—supine angle of trunk rotation lower, PO—pelvic obliquity, CRS—sum of cervical rotation to the left and right, LSF—sum of shoulder flexion limitation, LEE—sum of elbow extension limitation, LHE—sum of hip extension limitation, LHF—sum of hip flexion with knee extension limitation in both hip joints, NFL—neck flexion in side lying on the left, NEL—neck extension in side lying on the left, SFL—right scapula forward in side lying on the left, SBL—right scapula backward in side lying on the left, PFL—right part of pelvis forward in side lying on the left, PBL—right part of pelvis backward in side lying on the left, NFR—neck flexion in side lying on the right, NER—neck extension in side lying on the right, SFR—left scapula forward in side lying on the right, SBR—left scapula backward in side lying on the right, PFR—left part of pelvis forward in side lying on the right, PBR—left part of pelvis backward in side lying on the right, RS—sum of R coefficients, Cobb—Cobb angle, N—newton, m—meter, kg—kilogram, NA—not analyzed and *p*—statistical significance. Significance: (*)—at the level of 0.01 < *p* < 0.05, (**)—at the level of 0.001 ≤ *p* ≤ 0.01.

## Data Availability

The data presented in this study are available on request from the corresponding author. The data are not publicly available due to privacy restrictions.

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
