# Peer review of "Motor Function of Children with SMA1 and SMA2 Depends on the Neck and Trunk Muscle Strength, Deformation of the Spine, and the Range of Motion in the Limb Joints"

_ijerph, 2021, doi:10.3390/ijerph18179134_

Round 1

Reviewer 1 Report

There are several problems in the manuscript:

  1. The title and the aim of the manuscript are not matching.  The title indicates that the study investigates the motor function of the children with SMA1 and SMA2 threated pharmacologically, however, the results include measurements at the initial exams before starting any type of treatments. There are no results following the completing any pharmacological treatments. The authors should not use this title if they don’t include the post pharmacological measurements.
  2. The range of the values should be included in the tables as (min-max).
  3. Relationship Tables (Table 2 and 4) include the same values twice for example MF- CRS and CRS-MF are both included in Table 2 . Repeated results should be deleted from both tables.
  4. It is very difficult to follow the result section. It would be very helpful to include a figure to show/ describe the outcome measures. Many abbreviations are used in the text and in the tables that the readers constantly stop and search for the meaning of that specific abbreviation. Key index at the beginning would be very helpful to address this problem.
  5. The legend of Figure 1 should include the main message, and the color codes.  
  6. Please report the values of the left and right sides of the same task in one row instead of in two rows For example: The values for NEL and NER should be in one row in which top value indicates the left and bottom value  indicates the right side. This can be applied to NFL & NFR, SFL & SFR etc.   
  7. The outcome measures should be grouped for the neck, trunk, upper and lower limbs functions instead of combining all in a big table.  
  8. The Discussion section should be expanded to include significant correlations in the results. For example, in line starting in 293, it says that “several” , this should be changed to show how many. Another example is in line 309, “ many”

“ In the HFMS group several negative strong  correlations were observed between the measurements of strength in the upper and lower trunk and contractures in elbow joints (Table 3).  

Participants showed many significant relationships between the ROM in the neck 309 and joints of the limbs, with more significant relationships in the CHOP group

  1. Line 4 should be corrected:

 “The purpose of the research was objective was to investigate the functional relationships “  

  1. Editing through the manuscript is needed. Numbers > 10 should be written as a number (such as in line 62)

Author Response

Dear Reviewer,

Thank you very much for all your valuable advice and suggestions. We responded to all comments. Thank you for your time and substantive support.

Reviewer 1

There are several problems in the manuscript:

  1. The title and the aim of the manuscript are not matching. The title indicates that the study investigates the motor function of the children with SMA1 and SMA2 threated pharmacologically, however, the results include measurements at the initial exams before starting any type of treatments. There are no results following the completing any pharmacological treatments. The authors should not use this title if they don’t include the post pharmacological measurements.

Thank you for the advice. The title of the article has been changed.

  1. The range of the values should be included in the tables as (min-max).

Min and max values ​​have been added in the tables. Line

  1. Relationship Tables (Table 2 and 4) include the same values twice for example MF- CRS and CRS-MF are both included in Table 2 . Repeated results should be deleted from both tables.

We have replaced Table 2 with Figures 2 and 3. Line 250-255.

Table 4 has been corrected.  Line 314.

  1. It is very difficult to follow the result section. It would be very helpful to include a figure to show/ describe the outcome measures. Many abbreviations are used in the text and in the tables that the readers constantly stop and search for the meaning of that specific abbreviation. Key index at the beginning would be very helpful to address this problem.

A figure with abbreviations used in the work has been added at the end of the Methods section. Line 122-127, 169-170.

We also added two figures in the results to improve presentation of results.  Line 250-255.

  1. The legend of Figure 1 should include the main message, and the color codes.

Figure (now Figure 4) has been corrected - main information was provided at the top and the colors were described. Additionally, in the caption under the figure there are additional explanations.

 Line 350-354

  1. Please report the values of the left and right sides of the same task in one row instead of in two rows For example: The values for NEL and NER should be in one row in which top value indicates the left and bottom value indicates the right side. This can be applied to NFL & NFR, SFL & SFR etc.

Due to the necessity to include the minimum and maximum values ​​as suggested by the second reviewer, we set the NFL & NFR in one horizontal line and assigned them values ​​according to your suggestion.  Line 212-223.

  1. The outcome measures should be grouped for the neck, trunk, upper and lower limbs functions instead of combining all in a big table.

We agree that Table 1 contained a lot of information and theoretically it could be divided into several smaller ones. However, the original version of the manuscript already contained 4 tables. Therefore, we have divided Table 1 into two smaller tables. In the first one (Table 1) we provided basic information about the participants, and in the second one (Table 2) we grouped the measurements according to the type of examination (postural parameters, ROM, motor function and muscle strength, taking into account the head, upper and lower trunk). It seems to us that the Table 2 is now more understandable. Would you agree to accept the table as it is?

Line 205-223.

  1. The Discussion section should be expanded to include significant correlations in the results. For example, in line starting in 293, it says that “several” , this should be changed to show how many. Another example is in line 309, “ many”.

“ In the HFMS group several negative strong  correlations were observed between the measurements of strength in the upper and lower trunk and contractures in elbow joints (Table 3). 

Participants showed many significant relationships between the ROM in the neck 309 and joints of the limbs, with more significant relationships in the CHOP group

              Thank you for your advice. It was corrected. Line 326,324, 334.  

              We expanded discussion to add more information related to results. 

              Lines 374-  375, 393-403, 418-420, 428-436, 461-462, 464-468.  

  1. Line 4 should be corrected:

            “The purpose of the research was objective was to investigate the functional relationships “ 

               Thank you, we didn’t noticed a mistake. The text has been corrected

  1. Editing through the manuscript is needed. Numbers > 10 should be written as a number (such as in line 62)

            It was corrected . Line 63

Thank you very much for you support.

Reviewer 2 Report

This is a niche but interesting topic. Please look at these points:

  1. Lines 97-98: "CHOP INTEND and HFMSE functional scales" It can be interesting to add this scale tables in the manuscript (section material and methods)
  2. Lines 38-41: "as well as some problems in cervical facet injuries", please look at these recent refs.  - Regional and experiential differences in surgeon preference for the treatment of cervical facet injuries: a case study survey with the AO Spine Cervical Classification Validation Group. Eur Spine J. 2021 Feb;30(2):517-523. doi: 10.1007/s00586-020-06535-z.   ---  Muscle atrophy after treatment with Halovest. Spine (Phila Pa 1976). 2005 Jan 1;30(1):E8-12. doi: 10.1097/01.brs.0000148996.02
  3. Lines 339-342: "... been shown that motor skills of children with SMA can be affected by the strength of certain groups of muscles in... with spine, thoracic and pelvic deformities was observed" At this point of the discussion, the biomechanics of the cervical spine should be considered. Please look at these 2 important refs.  The Y-shaped trabecular bone structure in the odontoid process of the axis: a CT scan study in 54 healthy subjects and biomechanical considerations. J Neurosurg Spine. 2019 Feb 1:1-8. doi: 10.3171/2018.9.SPINE18396.   ----    Cervical rotation, chest deformity and pelvic obliquity in patients with spinal muscular atrophy. BMC Musculoskelet Disord. 2020 Nov 7;21(1):726. doi: 10.1186/s12891-020-03710
  4. Lines 360-367: "In the past, it has been shown that movement abilities may depend on the ROM in joints. In participants with SMA2 and SMA3, Salazar et al. observed that slight ROM limitation in the hip and knee joints may affect motor abilities... LHF ranges. More fit children showed smaller hamstrings contractures" These sentences should be merged to express a more linear concept. Please revise.
  5. English language requires some revision, as e.g. "PARTECIPANTS SHOWED many significant relationships between the ROM in the neck and joints of the limbs" (line 309); "In the literature, ONE CAN FIND publications on the limitations of the ranges of motion 410
    in people with SMA (line 410)"
  6. Lines 74-76: "The main objective was to investigate the functional relationships between selected ranges of motion of the neck, upper and lower limbs, the strength of the neck and trunk muscles, postural parameters and the MF of children with SMA1 and SMA2." However, the conclusion section (lines 437-446) is very general and it does not report the several results of this study. Please improve with more details.
  7. Figure 1 is very nice, but it should be discuss more in the figure legend.
  8. Another limitation of the study may be that drugs currently in use by these patients are different and not reported.

Overall a good paper.

Author Response

Dear Reviewer,

Thank you very much for all your valuable advice and suggestions. We responded to all comments. Thank you for your time and substantive support.

This is a niche but interesting topic. Please look at these points:

  1. Lines 97-98: "CHOP INTEND and HFMSE functional scales" It can be interesting to add this scale tables in the manuscript (section material and methods).

We know that adding these scales would make the text easier to read and understand. Unfortunately, both scales are extensive and their content exceeds the volume and purpose of this manuscript. Each scale consists of several pages of text. Therefore, we have added a text describing the study in general.  Line 99-112.

  1. Lines 38-41: "as well as some problems in cervical facet injuries", please look at these recent refs. - Regional and experiential differences in surgeon preference for the treatment of cervical facet injuries: a case study survey with the AO Spine Cervical Classification Validation Group. Eur Spine J. 2021 Feb;30(2):517-523. doi: 10.1007/s00586-020-06535-z.   ---  Muscle atrophy after treatment with Halovest. Spine (Phila Pa 1976). 2005 Jan 1;30(1):E8-12. doi: 10.1097/01.brs.0000148996.02

Thank you for pointing out very interesting references. We think there are no reports of cervical injuries in people with SMA, so we cant add this information in the symptoms. The articles were cited in the discussion. We added comments about the cervical spine .

Line 40, 469-481.

  1. Lines 339-342: "... been shown that motor skills of children with SMA can be affected by the strength of certain groups of muscles in... with spine, thoracic and pelvic deformities was observed" At this point of the discussion, the biomechanics of the cervical spine should be considered. Please look at these 2 important refs. The Y-shaped trabecular bone structure in the odontoid process of the axis: a CT scan study in 54 healthy subjects and biomechanical considerations. J Neurosurg Spine. 2019 Feb 1:1-8. doi: 10.3171/2018.9.SPINE18396.   ----    Cervical rotation, chest deformity and pelvic obliquity in patients with spinal muscular atrophy. BMC Musculoskelet Disord. 2020 Nov 7;21(1):726. doi: 10.1186/s12891-020-03710

The cervical biomechanics has been discussed in lines 469-481.  

We used the second reference “Cervical rotation, chest deformity and pelvic obliquity in patients with spinal muscular atrophy” as citations in line 441-447.  

Due to the lack of relationship between the publication “ The Y-shaped trabecular bone structure in the odontoid process of the axis: a CT scan study in 54 healthy subjects and biomechanical considerations. J Neurosurg Spine. 2019” and the topic of the presented study, we have omitted this article.

  1. Lines 360-367: "In the past, it has been shown that movement abilities may depend on the ROM in joints. In participants with SMA2 and SMA3, Salazar et al. observed that slight ROM limitation in the hip and knee joints may affect motor abilities... LHF ranges. More fit children showed smaller hamstrings contractures" These sentences should be merged to express a more linear concept. Please revise.

Thanks for your suggestion. The text has been completed to emphasize the concept.

Line  411-414.

  1. English language requires some revision, as e.g. "PARTECIPANTS SHOWED many significant relationships between the ROM in the neck and joints of the limbs" (line 309); "In the literature, ONE CAN FIND publications on the limitations of the ranges of motion 410in people with SMA (line 410)"

It was corrected.

  1. Lines 74-76: "The main objective was to investigate the functional relationships between selected ranges of motion of the neck, upper and lower limbs, the strength of the neck and trunk muscles, postural parameters and the MF of children with SMA1 and SMA2." However, the conclusion section (lines 437-446) is very general and it does not report the several results of this study. Please improve with more details.

We tried to describe the results in detail in the Results section, but after your feedback we completed the conclusions. Line 518-535.

  1. Figure 1 is very nice, but it should be discuss more in the figure legend.

Figure (now Figure 2) has been corrected - main information was provided at the top and the colors were described. Additionally, in the caption under the figure there are additional explanations. Line 350-354.

  1. Another limitation of the study may be that drugs currently in use by these patients are different and not reported.

We added this limitation in discussion . Line  516-517.

Thank you very much for you support.

Round 2

Reviewer 2 Report

Authors solved all my criticisms.